# Histopathology and immunohistochemistry as prognostic factors for poorly differentiated thyroid cancer in a series of Polish patients

Agnieszka Walczyk[1,2]*, Janusz Kopczyński[3], Danuta Gąsior-Perczak[1,2], Iwona Pałyga[1,2], Artur Kowalik[4], Magdalena Chrapek[5], Maria Hejnold[3], Stanisław Góźdź[2,6], Aldona Kowalska[1,2]

1 Endocrinology Clinic, Holycross Cancer Center, Kielce, Poland, 2 Collegium Medicum, Jan Kochanowski University, Kielce, Poland, 3 Department of Surgical Pathology, Holycross Cancer Center, Kielce, Poland, 4 Department of Molecular Diagnostics, Holycross Cancer Center, Kielce, Poland, 5 Faculty of Natural Sciences, Jan Kochanowski University, Kielce, Poland, 6 Clinical Oncology Clinic, Holycross Cancer Center, Kielce, Poland

* a.walczyk@post.pl

## Abstract

### Background

Poorly differentiated thyroid cancer (PDTC) is a rare but aggressive type of thyroid cancer (TC) and the main cause of death from non-anaplastic follicular cell-derived TC. Although the Turin criteria are well defined, the pathological features that could serve as diagnostic and prognostic factors remain controversial.

### Materials and methods

Forty-nine consecutive PDTC cases were identified in a single cancer center between 2000 and 2018. We analyzed the impact of routine histopathological and immunohistochemical features and several parameters that are not routinely included in pathology reports such as the presence of atypical mitoses, the amount of necrosis, or insulin-like growth factor-II mRNA-binding protein 3 immunostaining on the survival of patients with PDTC. Overall survival (OS) and disease-specific survival (DSS) were calculated using the Kaplan-Meier method.

### Results

Of the 49 PDTC 34 (69.4%) showed the insular pattern of growth. The median of poorly differentiated area was 95% (range, 1–100), and 30 (61.2%) patients had a predominant (>50%) insular area. The 5-year OS and DSS rates at a median follow-up of 57 months were 60.6% and 64.3%, respectively. Univariate analysis showed that tumor size >4 cm, presence of atypical mitoses, Ki-67 >5%, and thyroglobulin (Tg)-negative immunostaining were associated with a higher risk of PDTC-related death. Atypical mitoses and Tg negativity were independent factors of worse DSS in multivariate analysis. Patients with insular and

**Data Availability Statement:** All relevant data are within the manuscript and the figure files.

**Funding:** Project financed under the program of the Minister of Science and Higher Education called "Regional Initiative of Excellence" in the years, project no 024/RID/2018/19, amount of financing 11 999 000.00 zł.

**Competing interests:** The authors have declared that no competing interests exist.

predominant insular areas showed a 3- and 6-fold higher risk of PDTC death when they displayed atypical mitoses.

## Conclusions

In PDTC, the presence of atypical mitoses may be helpful in identifying patients with poorer outcome and worth including in pathology reports, particularly in tumors with a dominant insular pattern of growth. Additionally, the inclusion of Tg immunostaining may be considered in a prognostic context, and not only as a diagnostic feature.

## Introduction

Since the seminal reports in the early 1980s by Sakamoto et al.[1] and Cargangiu et al. [2] indicating that poorly differentiated thyroid cancer (PDTC) should be considered a distinct entity with an intermediate prognosis between differentiated thyroid cancer (TC) and anaplastic TC, studies focusing on this type of TC remain limited because of its relatively low incidence [3,4]. The reported prevalence rates vary from <1% of TCs to 6.7% depending on geographical regions [5,6]. It is likely that some genetic or environmental factors (including diet, iodine supply of studied population) may have an impact on the prevalence of PDTC in different areas [3,6]. Despite its rarity, PDTC is a clinically significant type of TC, as it is the main cause of death from non-anaplastic follicular cell-derived TC [3,6–10].

In 2004, the World Health Organization (WHO) recognized PDTC as a distinct entity among malignant thyroid tumors [11]. The recent 2017 WHO Classification of Endocrine Tumors defines PDTC as 'a follicular cell neoplasm that shows limited evidence of follicular cell differentiation and is morphologically and behaviorally intermediate between differentiated (papillary or follicular) carcinomas and anaplastic carcinoma' [12]. Because the diagnosis of PDTC is crucial for prognosis, the WHO definition is still relatively imprecise. It was therefore supplemented by a histopathological diagnostic algorithm termed the Turin criteria [13].

Based on architectural and high-grade features defined by the WHO and the Turin consensus, PDTC represents a heterogeneous group of malignant tumors originating from thyroid follicular cells. Hiltzik et al. [14] at Memorial Sloan Kettering Cancer Center (MSKCC) created the MSKCC-PDTC criteria to define PDTC according to the presence of mitosis and necrosis, and reported that PDTC is a homogenous and aggressive type of tumor in which growth patterns do not affect prognosis. Despite differences in the histologic definition of PDTC among pathologists, the recent 2017 WHO classification of endocrine tumors [12] does not recommend a diagnosis of PDTC in thyroid tumors that meet the MSKCC criteria, but do not meet the Turin criteria.

There is a universal agreement that proliferative grading (e.g., mitotic count or necrosis) should be described in pathology reports; however, the inclusion of non-routine parameters, such as the amount of necrosis or presence of atypical mitoses, remains debatable. The diagnosis of PDTC relies on histological features, and immunohistochemistry (IHC) is not required for diagnosis, although it may be useful. Recently, Asioli et al.[6] proposed the inclusion of a new prognostic factor in PDTC, namely, insulin-like growth factor-II mRNA-binding protein 3 (IMP3) expression, which can be evaluated by IHC.

The objective of the current study was to evaluate the impact of standard histopathological (HP) and immunohistochemical (IHC) features of PDTC, and several non-routine pathological parameters on the survival of PDTC-patients. IMP3 expression was included in the IHC

analysis to validate its power as a prognostic marker for PDTC in routine clinical practice as a first study beyond the original description by Asioli et al. [6].

## Materials and methods

### Study design and patients

All study procedures were approved by the Ethics Committee (EC) of Holycross Chamber of Physicians in Kielce, Poland. All the patients' data were fully anonymized before being accessed and leading to a review and subsequent analyses. Due to this fact, the EC waived the requirement for informed consent of the included patients.

The retrospective study was performed at Holycross Cancer Center (HCC), Kielce, a tertiary referral oncologic center in Poland, that provides comprehensive care for patients with TC and includes surgery, endocrinology, nuclear medicine, and radiation oncology departments, with similar characteristic as described in a previous study site [15]. The recruitment procedure resulting in a study group of 46 consecutive PDTC patients who underwent primary surgery between 2000 and 2017 was reported in details previously in the study with other objectives [15]. Subsequently, three PDTC cases were identified among 234 newly diagnosed TCs at HCC in 2018. Finally, the study group consisted of 49 consecutive PDTC patients retrieved in 2000–2018 among 2579 patients with TC diagnosed and treated in a single institution. The patients' medical records were reviewed, all data related to the objectives of the study were analyzed. The initial risk stratification was estimated according to the 2015 American Thyroid Association (ATA) modified initial risk stratification system [16]. The staging of all cases was re-classified according to the 8th edition of the American Joint Committee on Cancer/Tumor-Node-Metastases (AJCC/TNM) staging system [17]. All details of the management protocols, including indications for postoperative treatment such as radioiodine (RAI) therapy, external beam radiation therapy (EBRT), or subsequent-line therapies [conventional chemotherapy (CHTH) and tyrosine kinase inhibitor (TKI) therapy], and follow-up protocols, were described previously [15]. Death of the patient was considered disease-related when hospital conclusions were unequivocal or a TC death certificate was available. In one case (1/49; 2.04%), death data were obtained from the Polish National Cancer Registry, which is an official Polish registry of the incidence and mortality of all cancers, because the patient discontinued follow-up at HCC before the study summary. All the remaining patients were actively monitored until their death or, if alive, to the last follow-up date on February 28, 2019.

### Pathological review

Archival pathological materials (hematoxylin and eosin-stained slides, and formalin-fixed/paraffin-embedded blocks) were available for all cases. The diagnosis of PDTC was confirmed by two pathologists (JK and MH; both blinded to all the clinical data) independently based on the Turin consensus criteria as follows: a diagnosis of cancer of follicular cell derivation according to conventional criteria and (1) a solid, trabecular, or insular growth pattern; (2) absence of the conventional nuclear features of papillary TC; and (3) at least one of the following three features: convoluted nuclei (i.e., de-differentiated nuclear features of papillary cancer), ≥3 mitoses per 10 high-power fields (×400), and tumor necrosis [13].

The following features were assessed: tumor size, pattern of poorly differentiated (PD) growth, proportion of the PD area in each tumor, presence of predominant (>50%) insular pattern of growth and of well differentiated (WD) components, mitotic count and the presence of atypical mitoses, presence of tumor necrosis and its extent, convoluted nuclei, oncocytic features, and extent of vascular invasion.

The mitotic rate was determined by counting the number of mitotic figures in 10 high-power fields (×400) from hematoxylin and eosin-stained histological sections using a microscope (Olympus AX 60, Tokyo, Japan). According to the well-known protocol [18], the first field in each tumor lesion was selected randomly, and the following fields were sampled systematically using a mesh. On average, 10,000 nuclear profiles were counted per tumor lesion. Anything other than the typical form of normal mitoses, including asymmetrical, lagged, ring, multipolar, and anaphase-bridge mitoses were recognized as atypical mitoses [19]. Fresh tumor necrosis was classified as absent, focal (≤5% of the tumor area), or extensive (>5% of the tumor area) [14]. An oncocytic variant was defined as a PD area containing >75% oncocytic cells [20]. Vascular invasion was defined according to the criteria of the Armed Forces Institute of Pathology fascicle regarding thyroid tumors [21] as the presence of four or more foci of vascular invasion [14].

## Immunohistochemistry

IHC analysis of thyroglobulin (Tg; 2H11/6E1 cocktail), cytokeratin-19 (CK-19; RCK108), thyroid transcription factor-1 (TTF-1; 8G7G3/1), p53 (DO7), Ki-67 (MIB1), and IMP3 (L523) (all from Dako, Carpinteria, CA, USA) was performed on PD areas according to standard automated IHC procedures (Autostainer Link 48; Dako).

IHC expression of Tg, CK-19, TTF-1, p53, and IMP3 was scored according to the intensity and extent of staining using a semi-quantitative scoring system. The criteria for intensity scoring were as follows according to the well-known protocol [6]: 0 for no staining; 1+ for weak staining; 2+ for moderate staining; and 3+ for strong staining. The criteria for scoring the extent of staining were as follows: 0 for no staining; 1+ for 1–25%; 2+ for 26–50%; 3+ for 51–75%, and 4+ for 76–100% of stained neoplastic cells. The two scores were cumulated, and staining was considered positive when the combined IHC score was >2, whereas a final score of ≤2 was considered negative.

Ki-67 was scored according to the percentage of tumor cells expressing the proliferation marker Ki-67. The proliferation index was calculated for each tumor lesion by counting the total number of tumor cell nuclear profiles and the number of Ki-67-positive nuclear profiles in randomly selected fields [18]. Sections incubated identically except for replacement of the primary antibody by normal mouse IgG served as negative controls. Sections of normal tissue known to be immunoreactive for a test case were used as positive controls.

All quantified measurements and IHC-stained specimens were assessed by two pathologists (JK and MH) blindly and independently. In the case of discrepancies, consensus was reached regarding the mean value, which was considered for further analysis.

## Statistical analysis

Categorical data were expressed as numbers and percentages, whereas numerical variables were presented as range, median, and interquartile range. Survival curves were created using the Kaplan-Meier method. Hazard ratios (HRs) with 95% confidence intervals (95% CIs) for univariate and multivariate analyses were calculated using the Cox proportional-hazards model. Multivariate analysis included variables that were significantly related to survival in the univariate analysis. A two tailed p-value <0.05 was considered statistically significant. All statistical analyses were performed using R (version 3.1.2; The R Foundation for Statistical Computing, Vienna, Austria) and Statistica [TIBCO Software Inc. (2017) Statistica (data analysis software system), version 13; http://statistica.io].

## Results

### Clinical characteristics

The prevalence of PDTC in the HCC database was 1.89% (49/2579). The clinicopathological characteristics are presented in Table 1. The slight female preponderance was noted with the F/M ratio 2.1:1. In one case (2.0%), the initial ATA risk remained unknown because of insufficient data to distinguish high risk from intermediate risk.

Of the 49 patients, 48 (98%) underwent surgery as follows: in 85.7% total or near-total thyroidectomy was performed, and 12.3% patients underwent partial resection if complete resection was not achieved intraoperatively. In one case (2.0%), intraoperative procedures allowed to obtain a diagnosis of PDTC due to a HP report of PDTC metastasis to the supraclavicular lymph nodes. A primary thyroid tumor was unresectable in that case, but metastatic lymph node associated with a tumor were partly removed. Despite this fact, all HP and IHC features other than tumor size, and the proportion of PD/WD areas in the primary tumor, were determined from retrieved PDTC tissues. Lymph node dissection (LND) related to primary thyroid surgery was performed in 31 (63.3%) cases, whereas 17 patients (34.7%) showed no evidence of primary LND. The one patient (2.0%) with the unresectable thyroid mass mentioned above underwent removal of supraclavicular nodes. Postoperative RAI therapy was administered in 39 (79.6%) patients. Of the 49 studied patients, 21 (42.9%) received EBRT, however five (10.2%) received EBRT as single postoperative therapy. One (2.0%) received EBRT alone as palliative therapy.

During the follow-up period, 24 patients (49%) required more than one RAI therapy course (2–9 courses; activities: 2700–5550 MBq), and eight patients (16.3%) received systemic therapy as follows: three (6.1%) received CHTH, four (8.7%) received CHTH, and TKI (lenvatinib or vandetanib due to patients' participation in clinical trials) therapy, and one (2.0%) received TKI (sorafenib) alone. Twenty-three patients (46.9%) died. Disease-specific death was documented in 20 cases (40.8%).

### Pathological parameters

The summary of the pathological review is presented in Table 1. The median tumor size was 4.55 cm (range, 1.4–13 cm), and 29 patients (59.2%) had a primary tumor measuring >4 cm. Solid, trabecular, and insular patterns of growth were identified in 12 (24.5%), 3 (6.1%) and 34 (69.4%) patients, respectively (Fig 1). The PD area was >50% in most tumors (38/49; 77.6%), and the median PD area was 95% (range: 1–100). Of the tumors analyzed, 24 (49%) had a WD component, although in 13 tumors (13/49; 26.5%), the WD component occupied <25% of the tumor. A predominant (≥50%) insular pattern of growth was detected in 30 tumors (61.2%). A mitotic count ≥3/10 high-power fields was observed in 15 (30.6%) cases, and atypical mitoses were found in 18 tumors (36.7%) (Figs 1A and 2A). Necrotic areas were present in 34 (69.4%) tumors (Fig 1C), and the median rate of necrosis was 1% (range: 0–50); necrosis was extensive in 13 cases (26.5%) (Fig 2B).

### Immunohistochemical analysis

IHC analyses were performed in 48 cases (48/49; 98%), as samples from one tumor collected in 2004 were unsatisfactory. According to the ethic statement described above, all the patients' follow-up data were analyzed anonymously. However, the analysis of the medical record of one excluded IHC patient concluded that this patient is still alive and meets the criteria for an excellent response after primary therapy and maintains it on the summary date. The results of IHC analyses are summarized in Table 2. Most cases showed positive thyroglobulin (Fig 2C)

**Table 1. Clinical and pathological characteristics of the 49 patients with poorly differentiated thyroid cancer at the presentation stage.**

| Feature; all cases (n = 49) | |
|---|---|
| **Median age**, y ($Q_1$–$Q_3$; range) | 63 (50–70; 15–85) |
| **Age ≥55 y**, n (%) | 32 (65.3) |
| **Gender**, n (%) | |
| Female | 33 (67.3) |
| Male | 16 (32.7) |
| **Median tumor size**, cm ($Q_1$–$Q_3$) | 4.55 (3.0–7.0) |
| Range | 1.4–13.0 |
| **Tumor size**, cm, n (%) | |
| ≤ 4 | 19 (38.8) |
| > 4 | 29 (59.2) |
| Unknown | 1 (2.0) |
| **ATA initial risk**, n (%) | |
| Intermediate | 26 (53.1) |
| High | 22 (44.9) |
| Insufficient data | 1 (2.0) |
| **TNM stage**, n (%) | |
| I–II | 41 (83.7) |
| III–IV | 8 (16.3) |
| **Tumor pattern of PD growth**, n (%) | |
| Solid | 12 (24.5) |
| Trabecular | 3 (6.1) |
| Insular | 34 (69.4)[a] |
| **PD area**, n (%) | |
| < 10% | 3 (6.1) |
| ≥ 10%–<50% | 7 (14.3) |
| > 50% | 38 (77.6) |
| Unknown | 1 (2.0) |
| **Median PD area**, % ($Q_1$–$Q_3$; range) | 95 (75–100; 1–100) |
| **Predominant (≥50%) insular pattern of growth**, n (%) | 30 (61.2) |
| **Presence of well differentiated components**, n (%) | 24 (49) |
| Papillary cancer ≤25%, n (%) | 3 (6.1) |
| Follicular cancer ≤25%, n (%) | 10 (20.4) |
| Unknown | 1 (2.0) |
| **Oncocytic variant**, n (%) | 2 (4.1) |
| **Presence of necrosis**, n (%) | 34 (69.4) |
| Median amount of necrosis, % ($Q_1$–$Q_3$; range) | 1 (0–10; 0–50) |
| Extensive necrosis (>5% of the tumor area), n (%) | 13 (26.5) |
| **Convoluted nuclei**, n (%) | 32 (65.3) |
| **Mitotic count ≥3/10 high-power fields**, n (%) | 15 (30.6) |
| **Atypical mitoses**, n (%) | 18 (36.7) |
| **Extensive vascular invasion (≥4 foci)**, n (%) | 15 (30.6) |

[a] in one case, insular pattern of growth was determined in PDTC metastatic node

Abbreviations: ATA, American Thyroid Association; TNM, tumor-node-metastasis; PD, poorly differentiated.

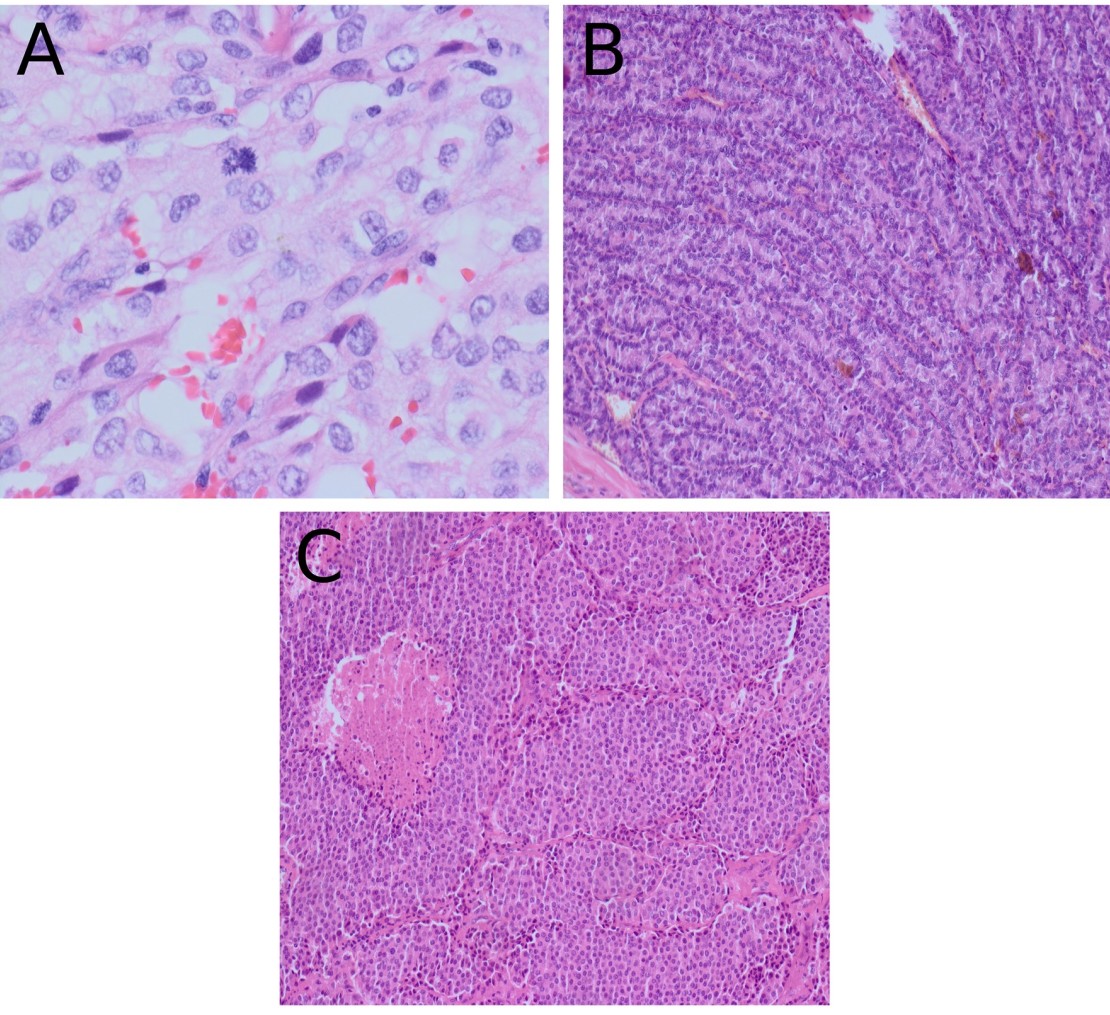

**Fig 1. Poorly differentiated thyroid cancer (PDTC) in H&E stain.** (A) PDTC with predominant solid growth pattern with atypical mitosis (×400), (B) trabecular pattern of growth (×200), and (C) insular pattern of growth with necrosis (×200).

and TTF-1 immunostaining (37/49; 77.1% and 42/49; 87.5%, respectively). Positive immunostaining for p53 and IMP3 was observed in 11 (23%) and 7 cases (14.6%), respectively. The median Ki-67 proliferative index was 5% (range, 1–80%).

## Survival analysis

The median follow-up was 57 months (range, 1–187 months; $Q_1$–$Q_3$, 30–101). The 5 and 10-year overall survival rates were 60.6% and 53%, and the disease-specific survival (DSS) rates were 64.3% and 56.3%, respectively. Seven patients (14.3%) died of the PDTC within 1 year from the diagnosis. This affected the mean values, as the 1 year DSS rate decreased to 85.4%.

Tumor size >4 cm, presence of atypical mitoses, Tg-negative immunostaining, and Ki-67 >5% were predictive factors of worse outcome based on univariate analysis, as shown in Table 3 and Fig 3. Because in most of the PDTC cases analyzed the PD area was >50% and the median PD area was 95%, univariate analysis was extended to PD threshold levels of ≥50%

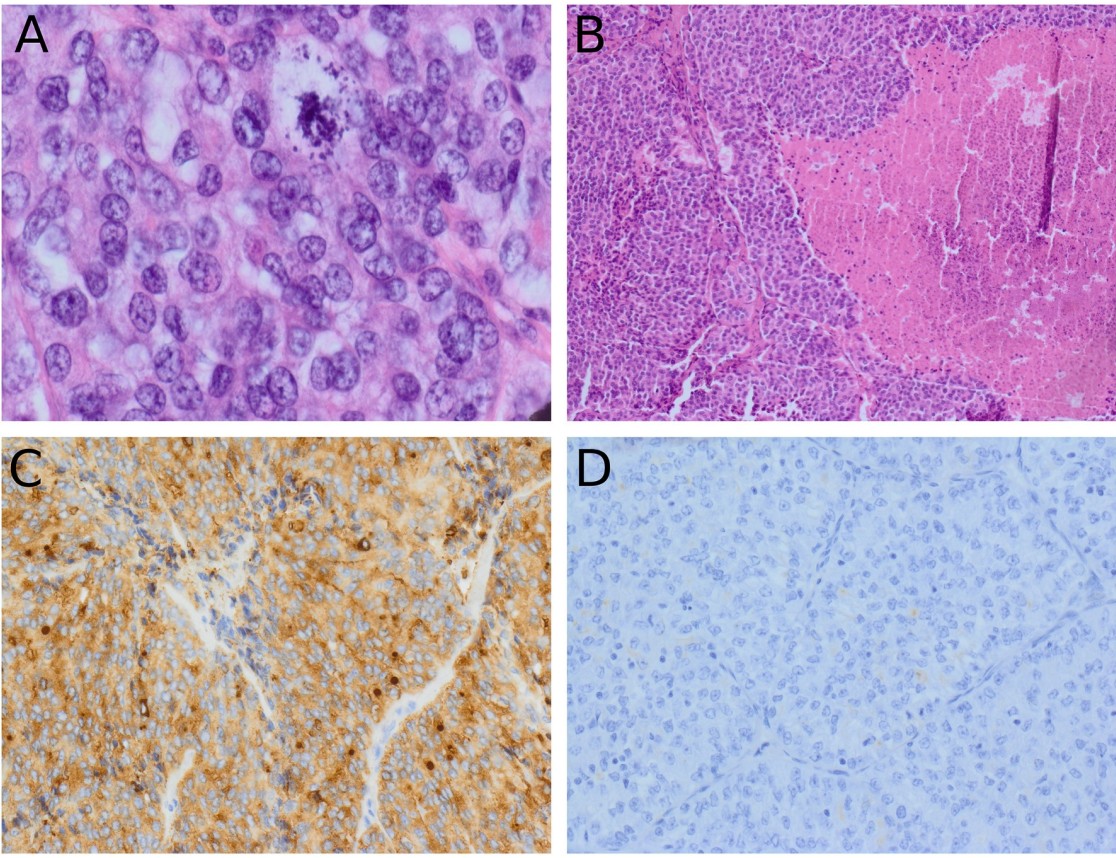

**Fig 2. Histological findings (H&E stain) in poorly differentiated thyroid cancer.** (A) Atypical mitosis in a tumor with a predominant insular pattern of growth (×400), and (B) extensive necrosis (×200). Thyroglobulin immunohistochemistry: (C) positive and (D) negative.

and 95% as predictive factors of DSS; however, there was no significant effect on survival. Patients with insular PDTC (n = 34) and those with a predominant insular area (n = 30) had a 3- and 6-fold higher risk of PDTC death, respectively, when they displayed atypical mitoses (Table 4). Multivariate analysis showed that only the presence of atypical mitoses and Tg negativity (Fig 2D) were independent factors of worse DSS (Table 5).

**Table 2. Immunohistochemistry in poorly differentiated thyroid cancer (n = 48).**

| Immunohistochemical feature | n (%) |
|---|---|
| **Thyroglobulin positive**, n (%) | 37 (77.1) |
| **CK-19 positive**, n (%) | 13 (27.1) |
| **TTF-1 positive**, n (%) | 42 (87.5) |
| **p53 positive**, n (%) | 11 (22.9) |
| **Ki-67**, %, median ($Q_1$–$Q_3$; range) | 5 (2–10; 1–80) |
| **IMP3 positive**, n (%) | 7 (14.6) |

Abbreviations: CK-19, cytokeratin-19; TTF-1, thyroid transcription factor-1; IMP3, insulin-like growth factor-II mRNA-binding protein 3.

**Table 3. Predictive factors of disease-specific survival in PDTC patients.**

| Variable | Characteristics of DSS | | Univariate analysis | |
|---|---|---|---|---|
| | 5 year DSS, % | 10 year DSS,% | HR [95% CI] | P |
| **Age,y** | | | | |
| < 55 | 63 | 63 | Ref.lev. | |
| ≥ 55 | 64 | 51 | 1.0 [0.4–2.6] | 0.93 |
| **Gender** | | | | |
| Female | 64 | 61 | Ref.lev. | |
| Male | 64 | 48 | 1.2 [0.5–3.0] | 0.73 |
| **Tumor size, cm** | | | | |
| ≤ 4 | 80 | 80 | Ref.lev. | |
| > 4 | 58 | 46 | 3.9 [1.1–13.3] | 0.033 |
| **Pattern of growth[a]** | | | | |
| Solid | 58 | 58 | 1.4[0.5–3.9] | 0.56 |
| Insular | 64 | 56 | Ref.lev. | |
| Trabecular | - | - | - | |
| **PD area** | | | | |
| ≤ 50% | 70 | 54 | Ref.lev. | |
| > 50% | 64 | 61 | 1.2[0.4–3.5] | 0.78 |
| **PD area** | | | | |
| ≤ 95% (median value) | 75 | 66 | Ref.lev. | |
| > 95% | 58 | 52 | 1.6 [0.6–4.1] | 0.31 |
| **Predominant insular pattern of growth** | | | | |
| No | 65 | 54 | Ref.lev. | |
| Yes | 65 | 60 | 1.1[0.4–2.7] | 0.91 |
| **Presence of WD component** | | | | |
| No | 58 | 52 | 1.6 [0.6–4.1] | 0.31 |
| Yes | 75 | 66 | Ref.lev. | |
| **Presence of necrosis** | | | | |
| No | 72 | 72 | Ref.lev. | |
| Yes | 62 | 52 | 1.7 [0.6–5.2] | 0.33 |
| **Amount of necrosis** | | | | |
| ≤ 1% (median value) | 62 | 62 | Ref.lev. | |
| > 1% | 67 | 51 | 1.4 [0.6–3.3] | 0.47 |
| **Extensive necrosis >5% of the tumor area, n (%)** | | | | |
| No | 66 | 56 | Ref.lev. | |
| Yes | 58 | 58 | 1.5 [0.6–3.8] | 0.43 |
| **Convoluted nuclei** | | | | |
| No | 63 | 46 | Ref.lev. | |
| Yes | 65 | 65 | 0.6 [0.3–1.5] | 0.29 |
| **Mitosis ≥3/10 high-power fields** | | | | |
| No | 68 | 62 | Ref.lev | |
| Yes | 57 | 46 | 1.7 [0.7–4.6] | 0.25 |
| **Atypical mitoses** | | | | |
| No | 78 | 69 | Ref.lev. | |
| Yes | 41 | 34 | 3.2 [1.3–7.8] | 0.01 |
| **Extensive vascular invasion** | | | | |
| No | 68 | 60 | Ref.lev. | |
| Yes | 57 | 48 | 1.6 [0.6–3.8] | 0.34 |

(*Continued*)

**Table 3.** (Continued)

| Variable | Characteristics of DSS | | Univariate analysis | |
|---|---|---|---|---|
| | 5 year DSS, % | 10 year DSS,% | HR [95% CI] | P |
| **Ki-67 >5%** | | | | |
| No | 71 | 63 | Ref.lev. | |
| Yes | 42 | 32 | 2.8 [1.1–7.2] | 0.03 |
| **TTF-1 positive** | | | | |
| No | 67 | 67 | Ref.lev. | |
| Yes | 63 | 53 | 0.9 [0.3–3.0] | 0.82 |
| **p53 positive** | | | | |
| No | 67 | 60 | Ref.lev. | |
| Yes | 51 | 40 | 1.9 [0.7–5.0] | 0.19 |
| **Thyroglobulin positive** | | | | |
| No | 41 | 41 | 3.3 [1.3–8.5] | 0.01 |
| Yes | 70 | 59 | Ref.lev. | |
| **CK-19 positive** | | | | |
| No | 65 | 53 | Ref.lev. | |
| Yes | 62 | 62 | 1.1 [0.4–2.8] | 0.88 |
| **IMP3 positive** | | | | |
| No | 62 | 62 | Ref.lev. | |
| Yes | 69 | 26 | 1.3 [0.4–3.8] | 0.69 |

[a] patients with a trabecular pattern of growth were excluded because the number was too low (n = 3) for reliable analysis.

Abbreviations: DSS, disease-specific survival; PD, poorly differentiated; WD, well differentiated; CK-19, cytokeratin-19; TTF-1, thyroid transcription factor-1; IMP3, insulin-like growth factor-II mRNA-binding protein 3; HR, hazard ratio; CI, confidence interval; Ref. lev., reference level.

## Discussion

Despite a rapid increase in the incidence of TC worldwide including in Poland, PDTC remains a rare type of TC [22–25]. The prevalence of PDTC has remained stable in Poland, accounting for 1.75% in 2010 [26] and 1.89% in the present study, which is consistent with data on PDTC [12]. Nevertheless, PDTC remains a challenge for pathologists and clinicians because of difficulties associated with the diagnostic process and its aggressive disease course, which requires intensive management and close follow-up [27].

The present study was a comprehensive analysis of several pathological features in a series of consecutive PDTC tumors identified using the Turin criteria to determine their prognostic value in routine clinical practice. According to the ATA guidelines [16], standard surgical pathology reports in TC should include a description of basic tumor features and additional information that may be helpful for determining prognosis. However, there is no clear statement regarding the specific HP or IHC findings that should be included because of their prognostic value in PDTC. The prognostic role of a larger tumor size or tumor necrosis was reported previously [6,28]. In the current study, patients with tumors >4 cm had an almost 4-fold higher risk of death than those with smaller tumors. This result was obtained in the univariate analysis (HR = 3.9; 95% CI: 1.1–13.3; p = 0.033), whereas in the multivariate analysis, the difference was not significant. A similar result was reported by Asioli et al. [6], who showed the prognostic value of a larger tumor size in univariate, but not in multivariate analysis.

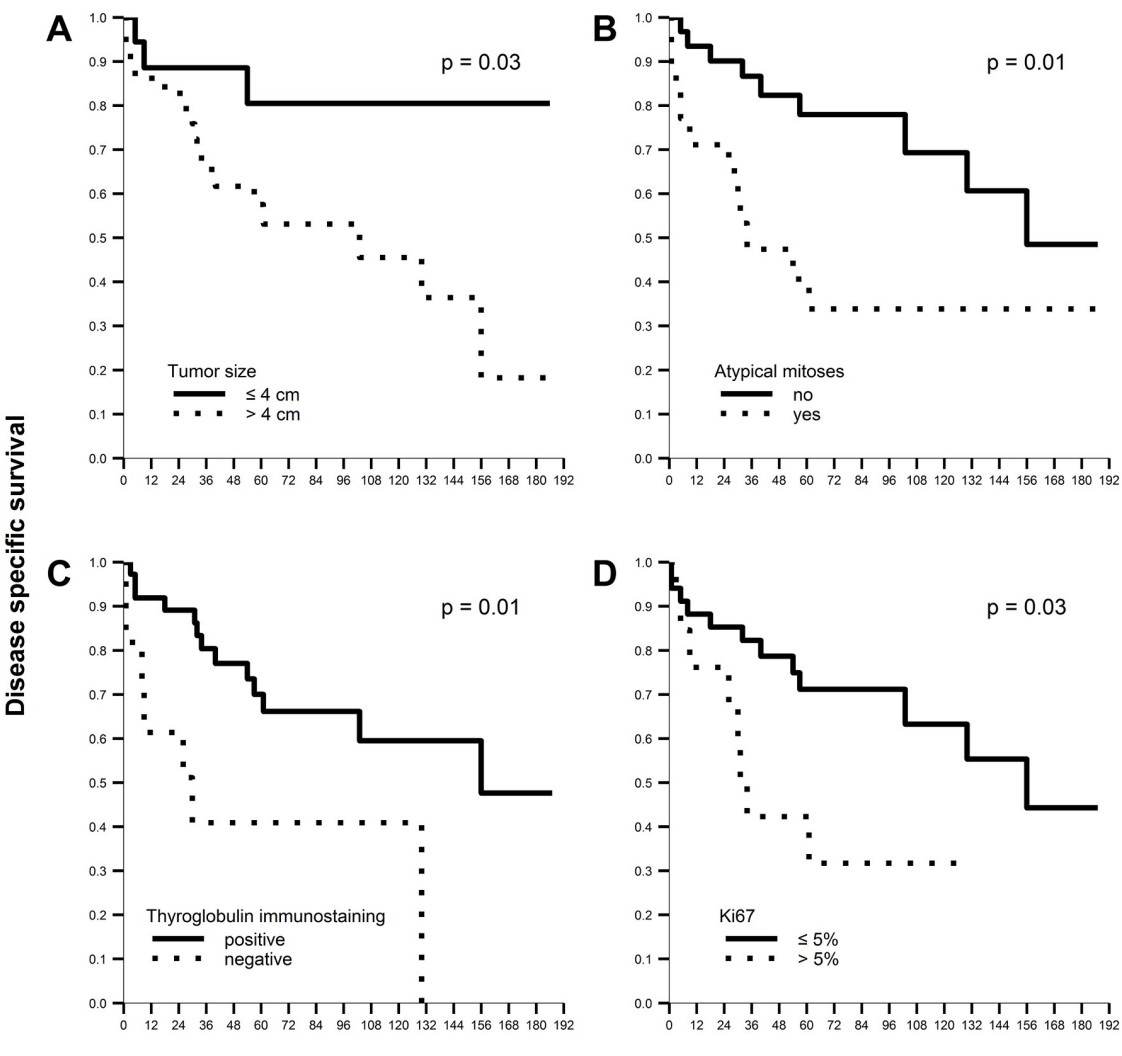

**Fig 3. Kaplan-Meier analysis of disease-specific survival in patients with poorly differentiated thyroid cancer.** According to (A) tumor size, (B) presence of atypical mitoses, (C) thyroglobulin-negative immunostaining, and (D) Ki-67 >5%.

It remains debatable whether the cut-off value for the PD area should be reported in PDTC tumors. The recent WHO classification [12] does not recommend using a cut-off value for the PD area to diagnose PDTC, and the prognostic implications of the proportion of PD area are equivocal. Volante et al. [29] reported that the extent of the PD area has no effect on overall survival in a large series of PDTC. Gnemmi et al. [28] reported no significant relation between the percentage of PD area and cancer-specific survival. Dettmer et al. [30] showed that even the presence of a small PD area of 10% had an effect on patient survival; however, Dettmer's study was based on a selected group of 42 PDTC cases with an adverse clinical outcome (recurrence or death). In the present study, which consisted of consecutive PDTC cases, a PD area was present in more than half of the tumor tissue in most patients, with a predominant insular pattern of growth and a high median PD area of 95%. These results suggest that the group of non-selective cases analyzed consisted of virtually 'pure' PDTC tumors. The low number of

**Table 4. The presence of atypical mitoses as a predictive factor of DSS in PDTC patients with an insular (n = 34), and a predominant (>50%) insular (n = 30), pattern of growth.**

| Variable | Characteristics of DSS | | | Univariate analysis | |
|---|---|---|---|---|---|
| | Patients, n (%) | 5 year DSS, % | 10 year DSS, % | HR [95% CI] | P |
| **Atypical mitoses in insular PDTC** (n = 34) | | | | | |
| No | 22 (64.7) | 80 | 70 | Ref.lev. | |
| Yes | 12 (35.3) | 44 | 33 | 3.0 [1.04–8.8] | 0.04 |
| **Atypical mitoses in PDTC with predominant insular pattern of growth** (n = 30) | | | | | |
| No | 19 (63.3) | 80 | 80 | Ref.lev. | |
| Yes | 11 (36.7) | 38 | NA[a] | 6.0 [1.5–23.5] | 0.01 |

[a] the longest follow-up was <10 years (76 months), and the patient is alive; the 10 year DSS could not be calculated.

Abbreviations: PDTC, poorly differentiated thyroid cancer; DSS, disease-specific survival; HR, hazard ratio; CI, confidence interval; Ref. lev., reference level.

PDTC cases with <10% of PD areas prevented a reliable analysis of 10% PD as a predictive factor of DSS. Nevertheless, we analyzed the effects of 50% and 95% (median value) as cut-off values, and the results showed no significant impact on DSS.

The prognostic significance of several high-grade features were reported in PDTC previously [6,14,28]. In the current study, only Ki-67 >5% and the presence of atypical mitoses correlated with decreased DSS in univariate analysis, whereas only atypical mitoses were an independent factor of worse outcome in multivariate analysis. This finding was unexpected because there are limited data [6,28] for determining the prognostic value of atypical mitoses in PDTC, and their prognostic significance in this type of TC has not been reported. Additionally, we analyzed the subgroups of insular pattern of growth and with predominant insular area, and found that patients in both subgroups had significantly shorter DSS, particularly when a dominant insular growth pattern coexisted with the presence of atypical mitoses. A recent review by Setia et al. [31] reported that, although atypical mitoses are present in PDTC, they are less common than in anaplastic TC. Atypical mitoses are also detected in other

**Table 5. Multivariate analysis of DSS in PDTC patients performed using the Cox proportional-hazards model.**

| Variable | Multivariate analysis | |
|---|---|---|
| | HR [95% CI] | P |
| **Tumor size, cm** | | |
| ≤ 4 | Ref.lev. | |
| > 4 | 3.3 [0.9–11.9] | 0.064 |
| **Atypical mitoses** | | |
| No | Ref.lev. | |
| Yes | 4.1 [1.5–11.0] | 0.005 |
| **Ki-67 >5%** | | |
| No | Ref.lev. | |
| Yes | 0.5 [0.1–2.5] | 0.40 |
| **Thyroglobulin positive** | | |
| No | 3.9 [1.3–11.3] | 0.014 |
| Yes | Ref.lev. | |

Abbreviations: DSS, disease-specific survival; PDTC, poorly differentiated thyroid cancer; HR, hazard ratio; CI, confidence interval; Ref. lev., reference level.

malignancies, although they are considered a prognostic factor only in pancreatic and breast cancers [19,32–34].

IMP3 detected by IHC was announced as a promising, cost-effective prognostic parameter; however, positive IMP3 immunostaining had no impact on patient survival in the present study, which is in disagreement with the results of Asioli et al. [6]. This result could be explained as follows: the number of IMP3-positive tumors was low in the group analyzed despite the use of the same methodology for IMP3 immunostaining recommended by Asioli for introduction into routine clinical practice; the Asioli's group was heterogeneous, and consisted of subgroups from two regions (the USA and Northern Italy) that differed significantly regarding the presence of an insular growth pattern, whereas the present group was homogenous and had a prominent insular pattern of growth. Finally, the impact of potential undefined population-dependent factors cannot be excluded. However, we found that convoluted nuclei were associated with a better prognosis according to Asioli's result [6], but in opposite to other report [28].

According to the WHO classification, a diagnosis of PDTC relies on histological features, and the presence of cell follicular differentiation. However, the use of an immunostaining panel may be supportive in difficult cases, e.g., Tg positivity can confirm an inconclusive diagnosis. Tg expression can be relatively decreased in PDTC and lost in anaplastic TC in parallel with thyroid cell de-differentiation from well-, *via* poorly-, to un-differentiated TC [6,13,35]. Although Tg negativity introduces a diagnostic challenge, it may be useful to predict a poor outcome in PDTC.

The present study had several limitations, including the small size of the study group and the retrospective design. The rarity of PDTC underlies the limited number of patients included, although 49 consecutive PDTC patients were analyzed. However, 49 patients were recruited from a 19-year database from one cancer center with consistent procedures for diagnosis, therapy protocols, and follow-up. Despite the fact that the study group is low, it may be exploitable for meta-analyses aimed for identification of the prognostic value of HP/IHC parameters in PDTC. In addition, IMP3 was assessed by IHC only. We did not perform a parallel molecular analysis in PDTC of 100% PD areas, as performed by Asioli et al. [6] to promote the use of IHC for IMP3 detection in routine clinical practice. Unfortunately, the present study does not provide any mutational status of PDTC cases, but the study group may constitute a base for further analyses of genetic alteration in PDTC of the Polish population. This would be highly advantageous to our knowledge-based approach with regards to the recent study on the genomic landscape of PDTC and its impact on the clinical outcome as reported by Landa et al. [36]. Meanwhile, clinicians may be supported by pathologists in identifying PDTC-patients with a higher risk of death. The surgery report of PDTC which is extended to the presence of atypical mitoses or Tg negativity in an IHC panel may be used as a potentially cost-effective diagnostic procedure of PDTC in cases of unknown genetic profiles in routine clinical practice.

## Conclusions

Standard surgical pathology reports in PDTC could be extended by the addition of several pathological parameters, which may be useful as prognostic factors. Further studies are still needed to validate the usefulness of IMP3 immunostaining for determining the prognosis of PDTC. The presence of atypical mitoses might be worth reporting, particularly in cases with a prominent insular pattern of growth. In addition, the inclusion of Tg immunostaining may be considered in a prognostic context, and not only as a diagnostic feature. However, all significant results of the current study need further analyses to confirm their potential prognostic value in the patients with PDTC.

## Author Contributions

**Conceptualization:** Agnieszka Walczyk, Aldona Kowalska.

**Data curation:** Agnieszka Walczyk, Janusz Kopczyński, Artur Kowalik, Magdalena Chrapek.

**Formal analysis:** Agnieszka Walczyk, Magdalena Chrapek.

**Investigation:** Agnieszka Walczyk, Janusz Kopczyński, Danuta Gąsior-Perczak, Iwona Pałyga, Maria Hejnold.

**Methodology:** Agnieszka Walczyk, Janusz Kopczyński, Artur Kowalik, Magdalena Chrapek, Maria Hejnold.

**Project administration:** Stanisław Góźdź, Aldona Kowalska.

**Resources:** Agnieszka Walczyk, Danuta Gąsior-Perczak, Iwona Pałyga, Aldona Kowalska.

**Supervision:** Stanisław Góźdź, Aldona Kowalska.

**Writing – original draft:** Agnieszka Walczyk, Janusz Kopczyński.

**Writing – review & editing:** Agnieszka Walczyk, Stanisław Góźdź, Aldona Kowalska.

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
