## [Decision Letter · Decision Letter 0]

16 Dec 2019

PONE-D-19-30397

Histopathology and Immunohistochemistry as Prognostic Factors for Poorly Differentiated Thyroid Cancer in a Series of Polish Patients

PLOS ONE

Dear Dr. Walczyk, 

Thank you for submitting your manuscript to PLOS ONE. After careful consideration, we feel that it has merit but does not fully meet PLOS ONE’s publication criteria as it currently stands. Therefore, we invite you to submit a revised version of the manuscript that addresses the points raised during the review process.

We would appreciate receiving your revised manuscript by Jan 30 2020 11:59PM. To enhance the reproducibility of your results, we recommend that if applicable you deposit your laboratory protocols in protocols.io, where a protocol can be assigned its own identifier (DOI) such that it can be cited independently in the future. For instructions see: http://journals.plos.org/plosone/s/submission-guidelines#loc-laboratory-protocols

We look forward to receiving your revised manuscript.

Kind regards,

Jason Chia-Hsun Hsieh, M.D. Ph.D

Academic Editor

PLOS ONE

Journal Requirements:

2. Please include your tables as part of your main manuscript and remove the individual files. Please note that supplementary tables (should remain/ be uploaded )as separate "supporting information" files

3. In the ethics statement in the manuscript and in the online submission form, please provide additional information about the patient records/samples used in your retrospective study. Specifically, please ensure that you have discussed whether all data/samples were fully anonymised before you accessed them and/or whether the IRB or ethics committee waived the requirement for informed consent. If patients provided informed written consent to have data/samples from their medical records used in research, please include this information.

Additionally, in regard to the following sentence: "IHC analyses were performed in 48 cases (48/49; 98%), as samples from one tumor collected in 2004 were unsatisfactory. However, the patient is still alive and meets the criteria for an excellent response after primary therapy and maintains it on the summary date.", please clarify whether patients' follow-up data were retrieved from anonymised medical records and authors did not have access to identifying information.

4. We noticed you have some minor occurrence(s) of overlapping text with the following previous publication(s), which needs to be addressed:

https://doi.org/10.1038/sj.bjc.6601453

https://doi.org/10.1038/modpathol.2010.117

https://doi.org/10.1111/cen.13910

In your revision ensure you cite all your sources (including your own works), and quote or rephrase any duplicated text outside the Methods section. Further consideration is dependent on these concerns being addressed.

Reviewers' comments:

Reviewer's Responses to Questions

**Comments to the Author**

1. Is the manuscript technically sound, and do the data support the conclusions?

Reviewer #1: Yes

Reviewer #2: Partly

2. Has the statistical analysis been performed appropriately and rigorously? 

Reviewer #1: Yes

Reviewer #2: No

3. Have the authors made all data underlying the findings in their manuscript fully available?

Reviewer #1: Yes

Reviewer #2: Yes

4. Is the manuscript presented in an intelligible fashion and written in standard English?

Reviewer #1: Yes

Reviewer #2: Yes

5. Review Comments to the Author

Reviewer #1: This manuscript describes a study where different IHC markers and histopathology features were associated with a poorer survival in a cohort of patients with Poorly Differentiated Thyroid Carcinoma (PDTC). As the authors state in the paper, PDTC, although rare, remains as the main cause of death from non-anaplastic follicular cell-derived thyroid cancers with a mean survival of 3.2 years, in part for a limited benefit from Radioiodine therapy. Therefore, there is an actual need for better diagnostic and prognostic markers. The authors conclude that in addition to the current recommendations by international guidelines for PDTC diagnosis and risk stratification, features such as the presence of atypical mitosis and negative Thyroglobulin (Tg) staining will be useful to identify those PDTC patients with a poorer outcome and worse prognosis.

Overall, this study is well conceived and executed. The manuscript is well written and the presented data support the conclusions.

Specific comments are listed that the reviewer hopes will improve the presentation:

1. As the authors already commented several times in the paper, the percentage of PDTC cases among all types of thyroid carcinomas may vary approximately from 1% to 7%. The series presented in this study represent a 1.89 % (49 out 2579 thyroid cancer cases in a single institution over a period of 18 years), which they claim is in concordance with other studies from Poland, the country where the institution is located. Previous papers published by the group of Dr. James A Fagin at MSKCC (Landa I, Ibrahimpasic T, et al., JCI 2016; Fagin JA, Wells SA, et al. N Eng J Med 2016), with data from the United States, state the PDTC prevalence rate is 6%. Can the authors describe what could be causing this difference and how the different pathological criteria (Turin vs MSKCC-PDTC definition) may affect this discrepancy?

2. This work doesn´t show any data on the mutational status of the studied samples. In the 2016 JCI paper from Landa I et al., it is shown that PDTCs represent an intermediate entity between Papillary and Anaplastic Carcinomas (PTCs and ATCS, respectively) regarding differentiation and mutational status. In this paper, PDTCs fulfilling the Turin definition were enriched in RAS mutations while MSKCC-PDTCs were enriched in BRAF mutations. Moreover, Landa et al. showed that mutational burden, EIF1AX mutations (co-ocurring with RAS mutations) and TERT promoter mutations – among other clinicopathological characteristics like gender or tumor size- were associated with survival in PDTC patients. The paper would gain a lot of relevance if the authors could provide mutational status of the referred genes. Even if authors are not able to deliver a genetic characterization, it would still be interesting to discuss this in the paper and how atypical mitosis or Tg staining may represent or not a more feasible and money-wise approach than sequencing in clinical practice.

Reviewer #2: This is an interesting article about using hisopathology and immunohistochemistry as prognostic factors for poorly differentiated thyroid cancer. The author concluded the presence of atypical mitoses may be predictor of mortality in patients with poorly differentiated thyroid cancer. The article has important clinical value due to rarity of poorly differentiated thyroid cancer patients. However, several questions should be made clear:

1. The sample size is relative small (n=49), please provide evidence of adequate power and sample size needed for testing main effect or interaction effect in this survival analysis.

2. Extrathyroid extension has been shown to related to mortality in patients with poorly differentiated patients. Please provided the % of extrathyroid extension and be included in multivariate analysis.

6. PLOS authors have the option to publish the peer review history of their article (what does this mean?). If published, this will include your full peer review and any attached files.

Reviewer #1: No

Reviewer #2: No

---

## [Author Response · Author response to Decision Letter 0]

21 Jan 2020

Academic Editor:

On behalf of myself and all co-authors, I would like to thank you for your revision and the constructive guidelines and comments, which will let us improve the manuscript to be accepted for the publication in PLOS ONE. 

Question 1. 

 The revised manuscript was improved according PLOS ONE’s style templates to meet the journal requirements which were presented in particular guidelines for Authors. 

Question 2. 

 All the tables were re-designed according to the PLOS ONE’s formatting rules and included to the revised manuscript after paragraphs in which they had been cited first. Additionally, all the figures captions were included in the relevant places of the revised manuscript, but the figure files were submitted in individual files in formats ‘.tiff’ according to the PLOS ONE’s formatting rules. 

Question 3. 

The ethics statement was extended to the statement that all the initial data as well as follow-up data were analysed anonymously as well as the ethics committee waived the requirement for informed consent (line 92-95). We also highlighted that pathologists were blinded to the clinical data and outcome (line 123). In regard to the sentence about results of IHC analyses we added again the clear statement that the follow-up data were analysed anonymously including a medical record of a patient without IHC analysis but with known clinical outcome (line 232-234). 

Question 4. 

Unfortunately, your HTML markup was not attached to your post. However, we scanned our manuscript by an overlapping program and finally we found few sentences in the Methods section and improved them. According to the cited by pathologists protocol of mitotic rate and Ki-67 in the Methods section [identified by you as https://doi.org/10.1038/sj.bjc.6601453] in line 136 and 162, we added a new reference in that paragraphs as 18th position. So, all the references cited after [REF18] were re-counted and ordered de novo. 

We also added REF 6 [identified by you as https://doi.org/10.1038/modpathol.2010.117] in line 153. We cited our own work [REF 15; identified by you as https://doi.org/10.1111/cen.13910], in the Methods section, line 99. Outside this section we rephrased the text of our previous work in lines 182-183, 191-196 in the Results section. We also found that the sentence in lines 139-140 in the Methods section requires revision and we rephrased it. 

The overlapping program did not indicate any more overlapping text outside the Methods section which should have been revised. 

The additional authors’ remark: in the revised manuscript an update in the funding statement was included, according to the updated Jan Kochanowski University in Kielce requirements. We also changed the affiliations of two co-authors (SG, AK) into the updated name: ‘Collegium Medicum’, Jan Kochanowski University, Kielce. 

Reviewer 1:

On behalf of myself and all co-authors, I would like to thank you for your revision of our manuscript and the constructive guidelines and comments, which will let us improve the manuscript to be accepted for the publication in PLOS ONE. 

Question 1. 

The incidence of PDTC is not reported as being similar to other different studies. The recent WHO classification of endocrine tumors [REF 12 in manuscript] indicates that the frequency of PDTC varies from less than 1% (e.g. in Japan) to 6.7% in Northern Italy. However, it is worth noting that the lower percentage of PDTC is usually reported in populations of sufficient iodine supply. Firstly, in 2007, Sanders et al. in their well-known study put our attention on the potential impact of environmental, e.g. related to dietary factors (including iodine), or some genetic factors on the occurrence of PDTC in the patients’ population from different areas. In Northern Italy, which was a region of a long-term insufficient supply of iodine, the occurrence was reported to be up to 15%, however, a later report showed a smaller proportion of PDTC, accounting for only 6.7% in a number of Italian patients [Asioli et al., 2010; REF 6 in manuscript]. Eventually, the WHO indicates a rate of 6.7% as being the highest reported figure from Europe. 

The Reviewer cites two studies based on PDTC patients from the USA, which is a leading country of sufficient iodine supplementation on a population level, and comments that the prevalence ratio was 6% from the US data. We would kindly like to remark, that in the first cited study published by Landa et al., we did not find any piece of information about the frequency of PDTC in the American population, despite the fact there was a very interesting analysis of the mutational status of 117 thyroid cancers containing 84 PDTC cases. In the second, by Fagin et al., we can indeed find the Authors’ note about the PDTC prevalence rate of 6%, but it is worth noting that the paper was a review, not an original study, and the Authors do not indicate any reference to the source of this fact. In summary, we do not find any evidence for such a high prevalence of 6% PDTC-patients from the USA, which is a country of long-term sufficient iodine supply. 

So far, the occurrence of PDTC in the USA, was reported as low as 1.8% (56/3128) in the original study by Asioli et al. from the well-known Mayo Clinic, in the USA, in which the Authors identified PDTC on the basis of the Turin criteria according to the recommendations by WHO in 2004 and 2018. The rate of 1.8% is indicated by the recent WHO classification as an evidenced rate of PDTC in patients from the USA. Additionally, we would like to note that Poland has been listed as an European country of sufficient iodine supply for close to 20 years after the introduction of the mandatory iodine prophylaxis program in 1997. This fact had been documented by the WHO in nationwide population-based studies performed in 2003 under the supervision of the International Council for the Control of Iodine Deficiency Disorders (ICCIDD; now Iodine Global Network [IGN]; http://ign.org/) and its Polish branch, and published by WHO in 2007 [Andersson M, de Benoist B et al. (eds) 2007 Iodine Deficiency in Europe, WHO]. In view of this data, our result of 1.89% of PDTC in Polish patients remains in accordance with the US data.

Despite the consideration above, we can agree with the Reviewer’s main remark, that the variation of PDTC incidence is an established fact, and the potentially causing factors have been added (line 57-59) to the revised manuscript. 

It is agreed that the pathologists can identify PDTC, however its histological definition is still under debate among pathologists. The use of the MSKCC criteria may probably better identify PDTC patients of high risk rather than the use of the Turin criteria. However, the recent WHO classification does not recommend a diagnosis of PDTC in thyroid tumors that meet the MSKCC criteria, but do not meet the Turin criteria. This statement is clearly written in the manuscript (line 74-76). The prevalence of PDTC in Poland is reported as based on the Turin criteria, according to the WHO recommendations. The MSKCC criteria was proposed on the basis of proliferate grading and that’s why they are known as more restrictive than the Turin criteria. The use of the MSKCC criteria will most likely decrease the amount of PDTC. The incidence of PDTC will be lower than the reported 1.89% (based on the Turin criteria) which are an expected result of the use of more restrictive pathological criteria with subsequent potential consequences being an unreliable analyses of very low subgroups. Even though, such analysis has not been performed, we would like to bring the Reviewer’s attention to the fact that the present study was designed on the basis of the present WHO definition of PDTC which still recommends the use of the Turin criteria. 

Question 2. 

Unfortunately, the mutational status of studied PDTC cases was not performed due to a financial aspect. We agree with the Reviewer’s remark, that such analysis would be very interesting, but it was not the aim of the current study. However, we consider it and we perform it willingly provided that the authors’ institution fund it. 

According to the Reviewer’s comment we agree that the potential clinical usefulness of a presence of atypical mitoses or Tg immunostaining in surgery report of PDTC should be more widely discussed. In our opinion this statement is more highlighted in the revised manuscript (lines 354-363), however a few comments were written in the primary version of the manuscript. 

Reviewer 2. 

On behalf of myself and all co-authors, I would like to thank you for your revision of our manuscript and the constructive guidelines and comments, which will let us improve the manuscript to be accepted for the publication in PLOS ONE. 

Question 1. 

We agree that the sample size is small. However, we mention that all subsequent patients with PDTC who came to our center between 2000 and 2018 were included, so it is not possible to increase the sample size to obtain the power of the tests at 80% level. Below we present the post-hoc calculations power of tests for the results included in Table 3. In that calculations, HR from Table 3 was taken as the assumed value of the effect. In addition, we also present the calculated sample sizes that would be necessary to achieve test power at 80%. All calculations were made using R statistical software with powerSurvEpi package.

Nevertheless, we mention that post-hoc calculations for test power (and especially the use of the observed effect value) are criticized by some authors (see e.g. https://journals.lww.com/annalsofsurgery/Citation/2019/01000/Don_t_Calculate_Post_hoc_Power_Using_Observed .46.aspx ).

Variable Variable

levels HR

(point estimate) Power of test

(post-hoc calculations) Sample size needed

for power of test = 80%

Age, yrs < 55 Ref. level 0.03 19665

 >=55 1.04 37016

Gender Female Ref. level 0.06 1703

 Male 1.2 826

Tumor size, cm <=4 Ref. level 0.63 29

 >4 3.9 44

Pattern of PD growth Solid 1.4 0.09 237

 Insular Ref. level 672

PD area <=50% Ref. level 0.05 867

 >50% 1.2 3292

PD area <=95% Ref. level 0.17 190

 >95% 1.6 190

Predominant insular pattern of growth No Ref. level 0.04 3639

 Yes 1.1 6065

Presence of WD component No 1.6 0.17 190

 Yes Ref. level 190

Presence of necrosis No Ref. level 0.17 121

 Yes 1.7 274

Amount of necrosis <=1% Ref. level 0.11 383

 >1% 1.4 312

Extensive necrosis > 5% No Ref. level 0.13 412

 Yes 1.5 149

Convoluted nuclei No 0.6 0.23 92

 Yes Ref. level 173

Mitosis >=3/10 high-power fields No Ref. level 0.21 201

 Yes 1.7 89

Atypical mitoses No Ref. level 0.72 38

 Yes 3.2 22

Extensive vascular invasion No Ref. level 0.20 220

 Yes 1.6 97

Ki-67 > 5% No Ref. level 0.15 47

 Yes 2.8 404

TTF-1 positive No Ref. level 0.04 1607

 Yes 0.9 11248

p53 positive No Ref. level 0.27 161

 Yes 1.9 48

Thyroglobulin positive No 3.3 0.73 14

 Yes Ref. level 44

CK-19 positive No Ref. level 0.04 6986

 Yes 1.1 2595

IMP3 positive No Ref. level 0.07 1333

 Yes 1.3 228

Based on https://acsjournals.onlinelibrary.wiley.com/doi/full/10.1002/cncr.29924 small, medium, and large HRs comparing 2 groups would be approximately 1.3, 1.9, and 2.8.

In the case of the results given in Table 3, in many cases the effect measured with HR is small (i.e. below 1.3) and only in 4 cases HR is greater or equal than 2.8. In all of this 4 cases (tumor size> 4, atypical mitoses, Ki-67> 5%, thyroglobulin positive) HR was statistically significant in our group of patients, and therefore our sample size was sufficient to confirm the occurrence of that large effects.

Question 2. 

 The majority of the present group (46/49) was recruited to the study focused on PDTC with other objectives [REF 15 in manuscript] which was designed and performed in our institution previously, as we described in the Methods section [line 100]. In that study the main aim was to validate the recent 8th Edition of AJCC/TNM staging system in a stratification of a risk of death of the patients with PDTC and evaluate the impact of the updated TNM stage on the patients’ survival. As we know, the major changes of the recent 8th Ed AJCC/TNM staging system comprised of an increase in the cut-off age from 45 to 55 years of age at diagnosis, and the removal of lymph nodes metastases and minor extrathyroidal extension (ETE) from the definition of tumor primary pT3 stage and resulted in a general trend towards the downstaging of TNM staging system. Hence, according to the aim of the previous study, the prognostic role of ETE in different aspects: presence or absence of ETE in PDTC, ETE extent (minor/gross) and their impact on PDTC-patients’ survival has been already analyzed in our 46 cases of PDTC and reported in detail previously [REF 15 in manuscript]. We agree that the presence/absence of ETE is an obligatory element in a surgery report of thyroid cancer, but its prognostic role was validated in the basis of our PDTC and an extension of the current analysis to 49 cases will not likely change the result, and it could be an overlapping analysis, so we decided not to perform it again. 

To summarise, we hope both the Reviewers and the Editorial Board will find the responses to the comments discussed above as satisfactory. 

Yours sincerely,

Agnieszka Walczyk

---

## [Decision Letter · Decision Letter 1]

4 Feb 2020

Histopathology and immunohistochemistry as prognostic factors for poorly differentiated thyroid cancer in a series of Polish patients

PONE-D-19-30397R1

Dear Dr. Walczyk,

We are pleased to inform you that your manuscript has been judged scientifically suitable for publication and will be formally accepted for publication once it complies with all outstanding technical requirements.

With kind regards,

Jason Chia-Hsun Hsieh, M.D. Ph.D

Academic Editor

PLOS ONE

Additional Editor Comments (optional):

All the questions were answered adequately.

Reviewers' comments:

Reviewer's Responses to Questions

**Comments to the Author**

1. If the authors have adequately addressed your comments raised in a previous round of review and you feel that this manuscript is now acceptable for publication, you may indicate that here to bypass the “Comments to the Author” section, enter your conflict of interest statement in the “Confidential to Editor” section, and submit your "Accept" recommendation.

Reviewer #1: All comments have been addressed

Reviewer #2: All comments have been addressed

2. Is the manuscript technically sound, and do the data support the conclusions?

Reviewer #1: Yes

Reviewer #2: Yes

3. Has the statistical analysis been performed appropriately and rigorously? 

Reviewer #1: Yes

Reviewer #2: Yes

4. Have the authors made all data underlying the findings in their manuscript fully available?

Reviewer #1: Yes

Reviewer #2: Yes

5. Is the manuscript presented in an intelligible fashion and written in standard English?

Reviewer #1: Yes

Reviewer #2: Yes

6. Review Comments to the Author

Reviewer #1: All my comments have been properly addressed by the authors in the rebuttal letter and the new versión of the manuscript as well.

Reviewer #2: Although whether or not violating the assumption of Cox proportional Hazard model is not well explained, but as the author state, the prevalence of PDTC is low, which make this article rare and naturally hard to be evaluated by traditional survival analysis assumptions.

7. PLOS authors have the option to publish the peer review history of their article (what does this mean?). If published, this will include your full peer review and any attached files.

Reviewer #1: No

Reviewer #2: No

---

## [Editor Report · Acceptance letter]

10 Feb 2020

PONE-D-19-30397R1 

Histopathology and immunohistochemistry as prognostic factors for poorly differentiated thyroid cancer in a series of Polish patients 

Dear Dr. Walczyk:

I am pleased to inform you that your manuscript has been deemed suitable for publication in PLOS ONE. Congratulations! Your manuscript is now with our production department. 

With kind regards,

on behalf of

Dr. Jason Chia-Hsun Hsieh 

Academic Editor

PLOS ONE